# Understanding Land-Use Trade-off Decision Making Using the Analytical Hierarchy Process: Insights from Agricultural Land Managers in Zambia

**Jane Musole Kwenye [1,\*], Xiaoting Hou Jones [2] and Alan Renwick [3]**

1   Department of Plant and Environmental Sciences, School of Natural Resources, Copperbelt University, Kitwe 10101, Zambia
2   International Institute for Environment and Development, London WC1V 7DN, UK
3   Centre of Excellence for Transformative Agribusiness, Faculty of Agribusiness and Commerce, Lincoln University, Lincoln 7647, New Zealand
\*   Correspondence: jane.kwenye@cbu.ac.zm

**Abstract:** Understanding factors that influence trade-offs between agricultural expansion and forest conservation is important in managing competing land-use objectives. This paper applies elements of the Analytical Hierarchy Process (AHP) to distinct farming business ownership models in Zambia to gain insights into factors that agricultural land managers take into account when considering land-use trade-off decisions which involve agricultural expansion into natural habitats. Results showed that the market domain was weighted above other domains, followed by the financial domain. When environmental considerations were pitted against other factors such as markets and finance, agricultural land managers were likely to trade off environmental concerns. Furthermore, availability of input subsidies via the agricultural Food Input Support Programme (FISP) influenced the decision to expand, particularly for the small-scale ownership model. This suggests that agricultural policies and strategies aimed at promoting agricultural productivity may require accompanying measures to safeguard forest ecosystems from agricultural expansion. Key differences in the factors influencing expansion decisions were evident among ownership models suggesting that ownership types do have an impact on factors considered. This highlights the need to advance tailored strategies that address differences in priorities and decision making emanating from variations in farming business ownership models.

**Keywords:** land-use trade-off; multi-criteria decision making; agricultural expansion; forest conservation; AHP





## 1. Introduction

The growing demand for food, driven largely by population growth, dietary shifts and economic growth [1–3], has been largely met by agricultural expansion at the expense of forest ecosystems in sub-Saharan Africa [4]. With the population of Zambia projected to more than double by 2050 [5], food demand is expected to escalate [3]. The demand for cereals, the country's staple food, is projected to rise to 519% of 2010 levels by 2050 [1]. Meeting the projected upswing in food demand requires increasing food production [6]. This can lead to competing land-use objectives for agricultural production and forest conservation considering that agricultural production is a major contributor to deforestation in Zambia [7] and other sub-Saharan African countries.

Agricultural expansion is projected to reduce 29% of the forest cover in sub-Saharan Africa by 2030 [8]. In Zambia, 90% of the deforestation in the country is due to agricultural expansion [7]. According to Zambia's Nationally Determined Contributions (NDCs), 8th National Development Plan, and the national strategy on Reducing Emissions from Deforestation and Forest Degradation (REDD+) [9,10], the country is committed to reducing

emissions from deforestation and forest degradation. However, the negative effects of agricultural expansion on forest cover conflict with this commitment. Therefore, trade-offs between competing land-use goals of preserving forests and expanding agricultural production to meet rising food demand must be recognized and managed.

Efforts to manage these trade-offs require an understanding of how those who own and manage land consider and manage difficult land-use trade-offs. Such an understanding can help to provide insights on how to negotiate the trade-offs. It can also help to influence those decision-making processes in view of developing sustainable agricultural development pathways for Zambia whilst providing lessons for other countries in sub-Saharan Africa. A possible way to gain insight into land-use trade-off decision making is to consider land managers who have made the decision to expand their agricultural area.

Past studies have shown that multiple factors are considered when making decisions on land-use change. For instance, factors such as biophysical, economic, technological, regulatory and personal characteristics of the land-use decision makers have been found to influence decisions on land-use change [11,12]. Using a case study of an irrigation scheme in New Zealand, it has been shown that financial, market, knowledge base, social well-being, environmental and regulatory factors are important in influencing land use transition [13]. Other studies report non-price factors, in particular, behavioral and psychological factors [14–19]. Therefore, approaches for examining land-use decision-making processes require considering the influence of these multiple factors [13].

A useful approach to addressing these multiple factors is within a Multi-Criteria Decision-Making (MCDM) Framework [13]. The MCDM framework is a potent analytical tool for assessing agricultural sustainability [20]. It is useful in agricultural and environmental land-use decision making to help identify the relative importance of trade-off factors (including economic, environmental and social factors) [21]. MCDM is also a useful framework for examining land-use decisions characterized by multiple and conflicting goals measured in disparate units [22,23]. A key advantage of an MCDM approach is its ability to enable a weighting of a range of selected criteria according to the individual's situation while integrating competing aspects of sustainability [12]. Thus, an MCDM approach is considered suitable for understanding agricultural decision making, since the method has the ability to consider trade-offs [21].

Against this background, this study applied elements of the MCDM framework—specifically, the AHP methodology—to test its applicability using distinct farming business ownership models in Zambia. The study sought to gain insights into factors that agricultural land managers with distinct ownership models consider in land-use trade-off decision making involving agricultural expansion into natural habitats. The study also sought to draw inferences on what this may mean for managing land-use trade-offs that consider the difficult task of balancing greater food production with the possible loss of ecosystem services.

Zambia provides a suitable context for undertaking this study for various reasons. By 2050, the country's population is projected to reach 39 million [5]. The rising population is projected to increase domestic food demand which is largely met through agricultural expansion at the expense of forests and biodiversity [24]. Despite the country's REDD+ commitments and targets to reduce the deforestation rate by 25% by 2025 [10,25], 90% of the deforestation in the country is driven by agricultural expansion [7]. This reveals the competing land-use objectives of increasing food production and forest conservation. It further shows the need for sufficient understanding of existing and future trade-offs between the competing objectives of producing more food and the conservation of forests [24]. This is fundamental given that managing these trade-offs remains poorly understood in African settings, notably in Zambia.

The paper proceeds with a description of the methods and materials followed by the results and discussion. The paper ends with conclusions and implications for managing land-use trade-offs involving the competing objectives of increasing food production and forest conservation.

## 2. Materials and Methods

### 2.1. Study Area

This research study was conducted in the Chongwe, Mkushi, Katete and Chipata districts situated in Lusaka, Central and Eastern Provinces of Zambia, respectively (Figure 1). The study sites were selected due to the presence of distinct farming business ownership models. The study sites are also situated in the same agro-ecological region which is the most productive agricultural region in the country. Because of this, the region is vulnerable to competing land-use goals involving agricultural expansion and forest conservation.

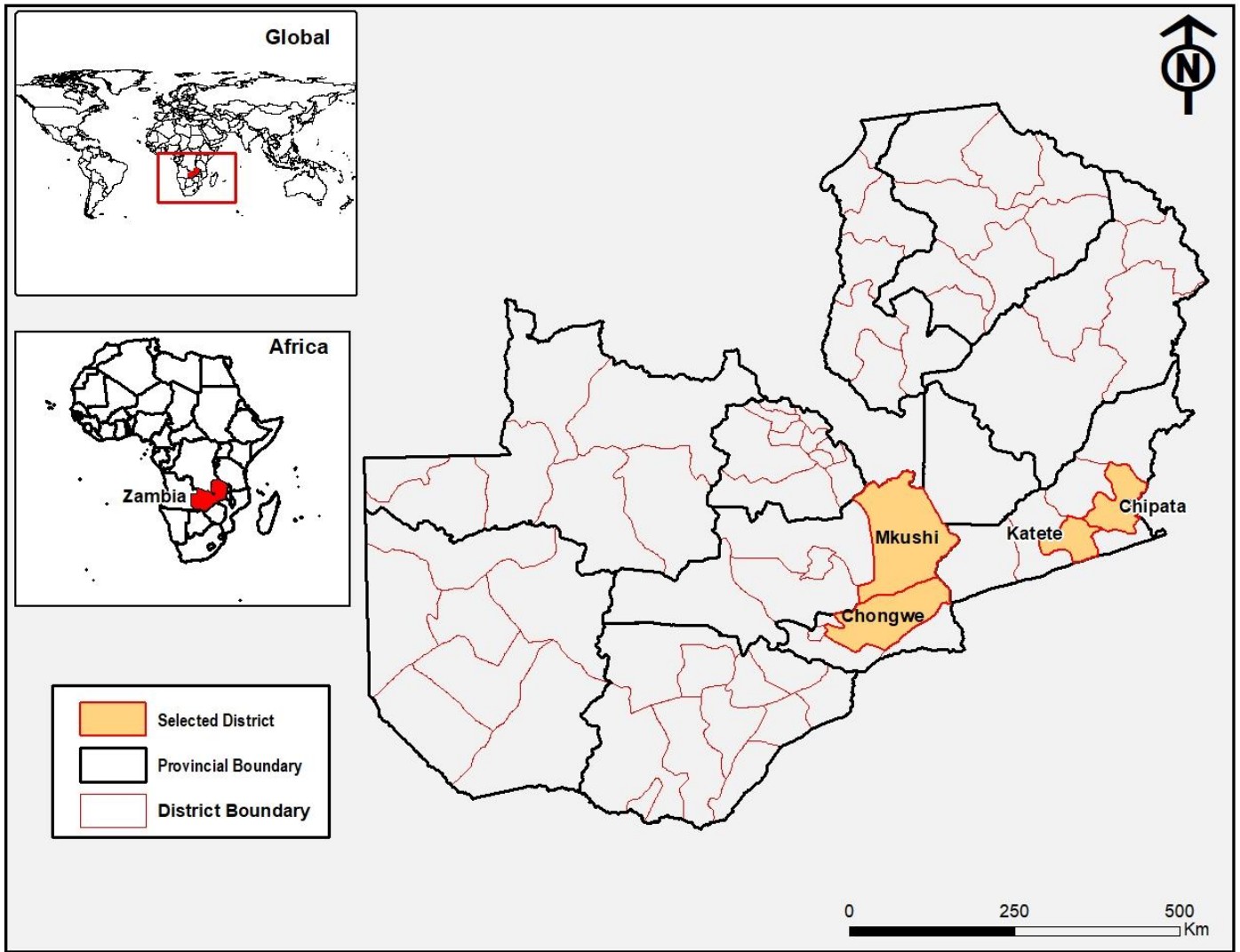

**Figure 1.** Map showing the study area.

Chongwe district covers an area of 2505 km$^2$. It is semi-rural and situated 50 km east of Lusaka city [26]. According to the most recent census, Chongwe district has a population of 313,386 people and is growing at a 2.4% annual rate [27]. Agriculture is the main economic activity in the district with maize, groundnuts and vegetables among the principal crops farmed in the area.

Mkushi district covers an area of 17,726 km$^2$ with an estimated population of 208,635 people and an annual population growth rate of 4.1% [27]. The district is characterized by large commercial agricultural activities. It hosts the majority of the country's commercial farms. Some crops that are commonly grown in this area include maize, soyabeans, beans, wheat and sunflower [28].

Katete district is situated 448 km east of Lusaka city, the country's capital. It shares an international border with Mozambique. The district covers an area of 2433 km$^2$ with an estimated population of 214,073 and an annual population growth rate of 2.4% [27]. It is predominantly rural with most of its population living in rural areas. Agriculture is the main economic activity in the district. Maize, cow peas, soya beans, groundnuts, sorghum, millet and sunflower are among the most commonly grown crops in the area [29].

Chipata district is located 600 km east of Lusaka city, the country's capital. It shares the border with Malawi and is about 110 km from its capital Lilongwe. The district covers an area of 6168 km$^2$ with an estimated population of 327,059 and an annual population growth rate of 2.8% [27]. The district's economy is anchored on agriculture. Major crops grown include maize, soya beans, millet, sorghum, cow peas, sunflower and groundnuts [30].

*2.2. Methods*

AHP Method and Application

A Multi-Criteria Decision-Making (MCDM) framework suggested by Renwick et al. [13] was used in this study. The MCDM framework was chosen for use in this study because it allows for the analysis of multiple domains in which alternatives are evaluated using a set of criteria, some of which are in conflict with one another [31]. The MCDM process also facilitates discussion of the decision-making process during its application [13]. This enables the generation of aggregate results while accounting for respondents' ideas and view points regarding the relative importance of the priorities [20,32]. Thus, it is suitable for questions concerning land-use trade-off decision making and has numerous applications in investigations of land-use choices [33].

In this study, we employed the MCDM to weight and rank factors that are fundamental to the land-use trade-off decision making of land managers with distinct farming business-ownership models. A range of methodologies has been devised to undertake MCDM analysis of which the MCDM itself is the general framework [34–36]. None of the several approaches can be emphatically regarded as superior to others given the multitude of applications for MCDM [13,37].

Within the MCDM general framework, this study employed elements of the Analytical Hierarchy Process (AHP) to assess tangible and intangible criteria in relative terms by means of an absolute scale [38,39]. Developed by Thomas Saaty in the late 1970s, the AHP method originates from the marketing sector [40]. It is relatively new in agricultural-related studies [20]. The AHP aids decision makers in constructing their preferences through criteria scoring and weighting [13] and is the most used MCDM method [20,41,42]. Since the current study involved understanding land-use trade-off decision making involving agricultural expansion into natural habitats, the AHP presented a suitable methodology for the study. The AHP methodology uses a pairwise comparison of selected criteria to demonstrate the relative importance of each criterion compared to other criteria [41–44]. This kind of analysis ensures an organized, rigorous, transparent and impartial evaluation of the options [31].

Through pairwise comparison, the AHP creates weights across the criteria that sum to 1 (or 100%) reflecting the overall decision-making process [13]. The generated weights for the criteria highlight how much influence the criteria have on the decision-making process. Thus, generated weights allow quantification of the overall importance of the criteria in the decision-making process. By generating weights for criteria that can be compared against one another, the AHP allows for evaluation of the relative importance of the criteria [33]. For instance, a criterion with a weight of 10% indicates that such a criterion has more influence on the decision-making process than a criterion with a weight of less than 10%.

Application of the MCDM, and the AHP in particular, requires identification of the criteria that are used to evaluate alternative systems [45,46]. Through a process involving a review of the literature, expert opinion and verification with those in land management, the key criteria (domains) for this study were established (Table 1). Sub-criteria for the domains were identified by study participants during data collection.

**Table 1.** Main domains for the MCDM Framework.

| Domain |
| --- |
| Market |
| Financial |
| Knowledge base |
| Social well-being |
| Regulation |
| Environment |

In the context of this study, drawing from past studies [13,33,45], the knowledge base domain refers to ease of access to technical knowledge and understanding of tools/equipment required by farming business ownership models. The market domain takes into consideration market availability and accessibility, and strength of the supply chain. The financial domain points to investment requirements and economic viability of crop production. The social wellbeing domain refers to improving the quality of life of farmers, consumers and members of the community and the capacity of farmers to produce enough food to feed communities. The regulation domain refers to availability and compliance to regulations and laws provided by public institutions that influence agricultural production and expansion. With regards to the environmental domain, this refers to improvement in the environmental quality of agricultural landscapes and consideration of negative environmental impact of agricultural expansion in its different forms.

*2.3. Data Collection and Analysis*

In-person interviews with agricultural land-use managers characterized by distinct farming business ownership models were conducted using the MCDM framework, AHP. While there are few farming business ownership models, these ownership forms adopt different decision-making processes and consider different factors when deciding how to manage agricultural expansion of their businesses. Therefore, to gain insights into the differences and identify any similarities, a range of farming business ownership models existing in Zambia were considered in this study. These include private individually owned (small, medium and commercial scale), Non-Governmental Organization (NGO)-owned, Government-owned and Group-owned (Cooperatives). Private individually owned farming business ownership models are managed by individual farmers. They are categorized by the scale of land holdings based on the classification provided by the Ministry of Agriculture in Zambia. The small-scale category refer to landholdings that are less than 5 hectares. The medium-scale and large-scale categories refer to landholdings that are 5–10 hectares and more than 10 hectares, respectively. Group-owned farming business ownership models are managed by a group of farmers commonly referred to as cooperatives. The NGO and Government farming business ownership models are managed by NGOs and the Government.

Study participants were purposively selected to represent the different farming business ownership models considered, as well as those who had previously decided to expand. Interviews were conducted on-site at the farms of the agricultural land managers from 1st to 20th December 2021. The farms were situated in the Mkushi, Chongwe, Katete and Chipata districts of Zambia. In total, seventeen (17) agricultural land managers were interviewed (Table 2).

The study participants represented the seven (7) farming business ownership models. Interview questions were framed in the context of what factors were important to them as land managers when deciding to expand their agricultural area. Interviews lasted on average one hour 30 min to two hours.

The AHP methodology was applied to participants at the main domain level and weights generated in terms of overall importance of each of the six domains in influencing the decision to expand. The interview-focused methodology sought to allow participants to determine the relative importance of factors considered when deciding to expand their

agricultural area. With the aid of a comparison scale (Table 3), participants performed pairwise comparisons.

**Table 2.** Characteristics of study sample.

| Interviewee No. | Farming Business Ownership Model | Agricultural System | Crops Grown | Farm Size (Ha) |
|---|---|---|---|---|
| 1 | Small-scale individually owned | Crop production | Maize, groundnuts, soya beans, sunflower | 4 |
| 2 | Small-scale individually owned | Crop production | Maize, tomatoes, soya beans, sunflower | 4 |
| 3 | Medium-scale individually owned | Crop production | Maize, cotton, groundnuts | 6 |
| 4 | Medium-scale individually owned | Crop production | Maize, groundnuts | 7 |
| 5 | Large-scale individually owned | Crop and fruit trees production | Maize, soya beans, ground nuts, sun flower, oranges | 50 |
| 6 | Large-scale individually owned | Crop production | Maize, soya beans, tomatoes, sunflower | 44 |
| 7 | Large-scale individually owned | Crop production | Maize, soya beans, rape, cucumber, watermelon cum | 40 |
| 8 | Government-owned | Crop production | Maize, sunflower, soya beans | 5000 |
| 9 | Government-owned | Crop production | Maize, wheat, soya beans | 750 |
| 10 | Government-owned | Crop production | Maize, wheat, soya beans | 2275 |
| 11 | Government-owned | Crop production | Maize, groundnuts, soya beans, sorghum, beans, wheat | 80 |
| 12 | Shareholder-owned | Crop production | Maize, tomatoes, onions, pepper, cabbage, wheat | 2900 |
| 13 | Shareholder-owned | Crop production | Maize, tomatoes, cabbage, lettuce herbs, carrots, beetroot, broccoli, cauliflower | 90 |
| 14 | Shareholder-owned | Crop production | Maize, wheat, soya beans | 63 |
| 15 | NGO-owned | Crop production | Maize, wheat, soya beans, beans, sorghum, sunflower, oats | 50 |
| 16 | Group-owned (Cooperative) | Crop production | Maize, soya beans | 5 |
| 17 | Group-owned (Cooperative) | Crop production | Maize, soya beans | 5 |

**Table 3.** Pairwise comparison scale.

| Level of Importance | Definition | Explanations |
|---|---|---|
| 1 | Equal importance | The two domains contribute equally to the decision process |
| 3 | Moderate importance | One domain is slightly more important than the other |
| 5 | Strong importance | One domain strongly dominates the other |
| 7 | Very strong importance | One domain very strongly dominates the other |
| 9 | Extreme importance | One domain completely dominates the other in the decision process |
| 2, 3, 6, 8 | Intermediate values | Expresses intermediate values |

Source: Saaty [39].

Pairwise comparisons for each participant was recorded using the Expert Choice software. During each comparison, participants were asked to express their preferences on a comparison scale that ranged from one (1) for criteria that were equally important to nine (9) for criteria that were significantly more important than the other. When setting their preferences, participants were asked to explain their choices. After the pairwise comparisons were completed, participants were shown their individual weights and had the option to adjust their choices if they were unsatisfied with the final weighting.

## 3. Results

Results for the main domains are presented first followed by those for the subdomains. To support quantitative results, context statements that emerged during the implementation of the MCDM framework are provided. This was possible given that the MCDM process facilitated discussion of the decision-making while the MCDM framework was being applied.

### 3.1. Relative Importance of the Main Domains in Influencing Land-Use Trade-off Decision Making

Summary statistics for each of the main domains across all farming business ownership models are presented in Table 4. Looking across the farming business ownership models, the market domain had the highest mean score followed by the financial domain. The environment domain had the lowest mean score.

**Table 4.** Summary statistics for the main domains across farming business ownership models.

| Domain | Mean | SD | Range | Max | Min |
|---|---|---|---|---|---|
| Market | 0.29 | 0.12 | 0.35 | 0.42 | 0.07 |
| Financial | 0.22 | 0.05 | 0.17 | 0.28 | 0.11 |
| Social well-being | 0.15 | 0.05 | 0.16 | 0.25 | 0.09 |
| Regulation | 0.14 | 0.08 | 0.18 | 0.24 | 0.06 |
| Knowledge base | 0.12 | 0.02 | 0.07 | 0.15 | 0.08 |
| Environment | 0.08 | 0.07 | 0.21 | 0.25 | 0.06 |

Derived weightings for each of the main domains for the farming business ownership models are presented in Figure 2a while Figure 2b presents the average across all the farming business ownership models. Despite managing agricultural land located in different geographical locations, thereby subject to different external drivers such as market forces among others, respondents placed high importance (weights) on the market domain (Figure 2a) relative to the other domains. This was observed for all the farming business ownership models except the NGO ownership model for which the environment domain dominated the decision-making process. The social well-being and market domains had higher importance in influencing the decision making for the government ownership model. When responses for all respondents were aggregated and an average taken (Figure 2b), the market domain was given higher weight followed by the financial domain. This demonstrates how important these domains are in influencing the decision to expand across the ownership models. The importance of the market domain in influencing the land-use trade-off decision-making process based on its weighting closely mirrored the views expressed by respondents during discussions.

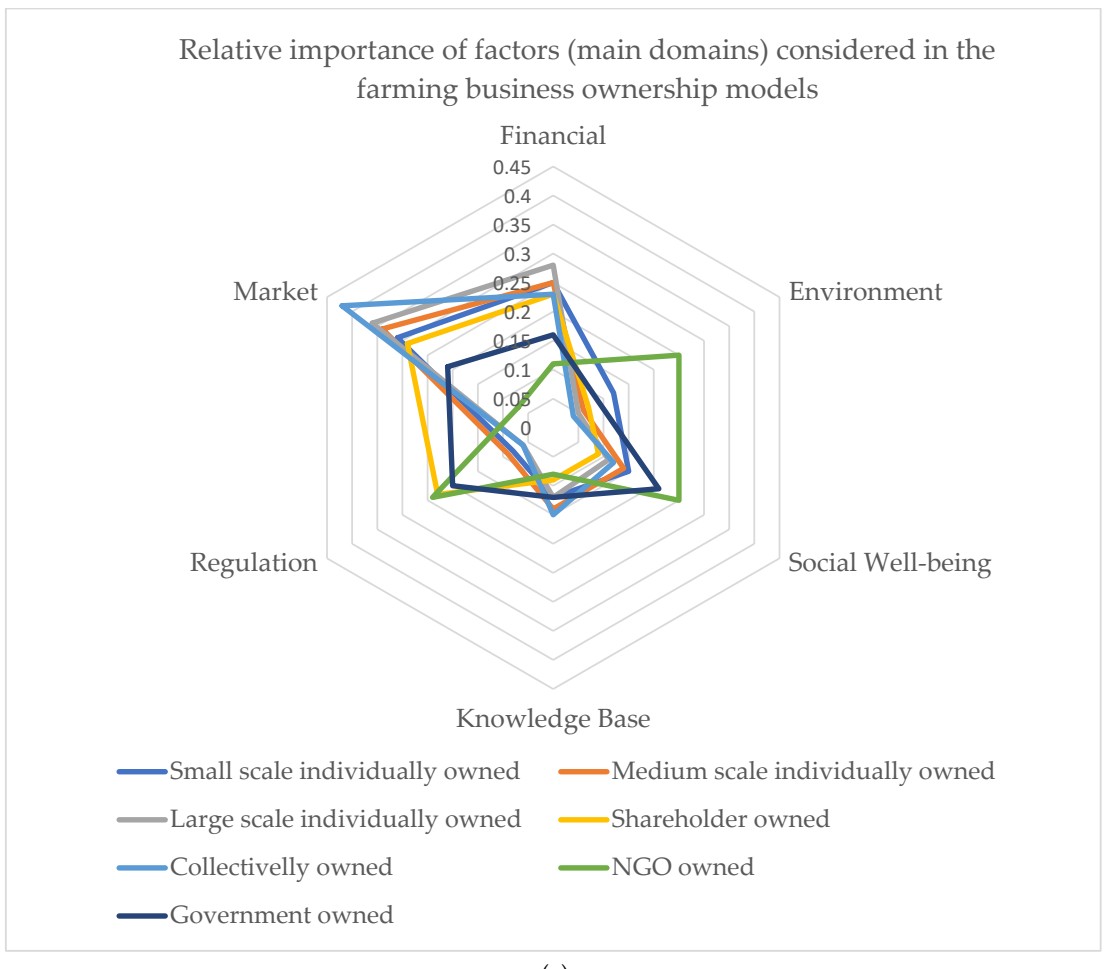

(**a**)

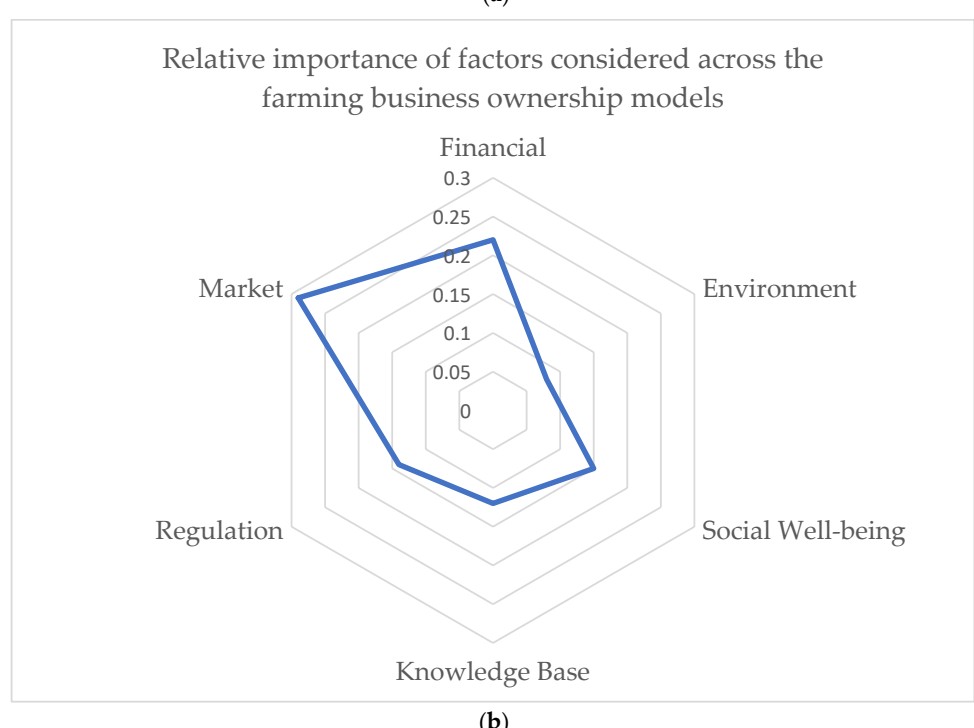

(**b**)

**Figure 2.** (**a**). Relative importance of the main domains in influencing land-use decision making for each farming ownership model. (**b**). Relative importance of main domains across all farming ownership models.

Shareholder-owned—"*Decisions on expansion are largely guided by the market…If there is nowhere to sell the produce then there is no need to expand crop production.*"

Government-owned—"*Market is very important . . . We can't grow something we cannot sell . . . As we plan to expand [the cropped area] we focus more on market availability for our produce*".

While the importance of the financial domain in influencing the land-use trade-off decision making was highlighted by the respondents, this was often tempered by the need to have a secure market for the produce in the first place.

Large-scale individually owned—"*Market for the produce comes first before anything else when considering expansion . . . We can't talk about profits before looking at the market.*"

Collectively owned—"*We can't think of profits before thinking of whether there is market for our produce . . . When planning expansion [of the area under cultivation] we think of where we will sell our produce first before we think of profits.*"

As highlighted in Table 3, on average, the social well-being domain was third in terms of its importance in influencing the decision to expand. This was supported by a range of comments that emerged during discussions.

Collectively owned—"*Financial and market factors are important however . . . we don't overlook the impact of our expansion on the quality of life of our members.*"

Large-scale individually owned—"*We don't overlook social well-being factors when planning to expand [the cultivated area] . . . we draw our labour force from the communities.*"

For the government ownership model, respondents considered social well-being factors to be more important than financial factors when making the decision whether or not to expand. This was reflected in the views expressed by the respondents.

Government-owned—"*Our aim is not to maximize profits when considering expansion [of the cropped area] because ours is a service . . . our mandate is ensure food security.*"

The importance of the social well-being domain in influencing decision making was also highlighted and emphasized by respondents who represented the collective ownership model. This was highlighted during the discussions.

Collectively owned—"*When expanding [the area under cultivation] our primary goal is to help improve the quality of life of the members . . . We work on ensuring that the expansion is beneficial to the members.*"

Looking across the ownership models, the regulation domain appeared to be, on average, of similar importance to the social well-being domain in influencing the decision to expand. Respondents, particularly those who represented shareholder, government and NGO ownership models indicated that this domain was an important element in the decision-making process.

Shareholder-owned—"*When considering expansion [of the cropped area] regulations have to be taken on board . . . Regulations will stop us from doing anything we want . . . We have to expand [the area under cultivation] within what the regulations stipulate*".

The MCDM results suggest that the environment domain had less importance in influencing the decision to expand with an average weight of only 8%. However, some respondents representing government and NGO ownership models prioritized environmentally friendly land-use decisions.

NGO-owned—"*The environment is important . . . We don't just make expansion [of cropped area] decisions based on our ability to sell . . . We also need to look at how we can protect the soil.*"

Some respondents gave examples of instances where they had foregone lucrative land-use options that were not environmentally friendly.

Government-owned—"*When we compromise our environment in the decision to expand [the cropped area] we affect sustainability of our production. Let's not forget the climate change part . . . If our expansion [of the cultivated area] is going to affect underground water systems or displace animal then we can't proceed with the expansion.*"

### 3.2. Subdomains Considered in Land-Use Trade-off Decision Making

The subdomains (criteria) for each of the main domains considered in the decision to expand were identified by respondents. Figure 3 and Table 5 present results on the subdomains that were taken into consideration. The number of subdomains considered ranged from 1 to 7. Relative to other domains, a higher number of subdomains was considered for the financial domain although it lagged behind the market domain in terms of influencing the decision to expand based on generated weights. Compared to other farming business ownership models, respondents of small-scale ownership models considered the highest number of subdomains for the financial domain, seven (7). While the shareholder and large-scale ownership models highlighted considerable subdomains for the environmental domain, on average, the weights generated for the associated main domains were lower compared to other ownership models. Overall, subdomains considered for the regulation domains were few across the farming business ownership with the small-scale and medium-scale ownership models each taking into account just one subdomain.

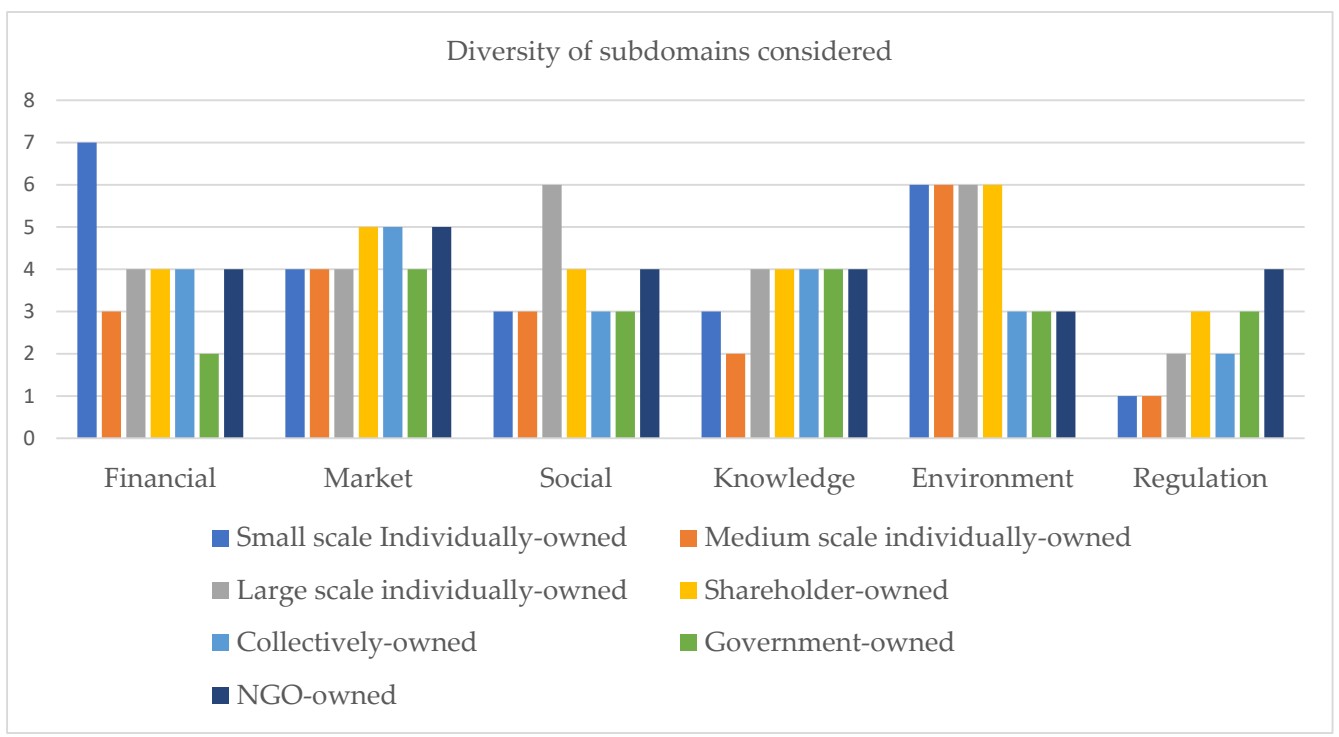

**Figure 3.** Diversity of subdomains considered across all respondents and farming business ownership models.

**Table 5.** Subdomains considered across farming business ownership models.

| Subdomains | Small-Scale Individually Owned | Medium-Scale Individually Owned | Large-Scale Individually Owned | Shareholder Owned | Collectively Owned | Government Owned | NGO Owned |
|---|---|---|---|---|---|---|---|
| Financial | | | | | | | |
| Capital/credit availability | ✓ | ✓ | ✓ | | ✓ | | |
| Profitability | ✓ | ✓ | ✓ | ✓ | ✓ | ✓ | ✓ |
| Payback period | ✓ | | ✓ | ✓ | ✓ | ✓ | |
| Profit variability | ✓ | ✓ | ✓ | ✓ | ✓ | | ✓ |

**Table 5.** *Cont.*

| Subdomains | Small-Scale Individually Owned | Medium-Scale Individually Owned | Large-Scale Individually Owned | Shareholder Owned | Collectively Owned | Government Owned | NGO Owned |
|---|---|---|---|---|---|---|---|
| Income diversification | √ | | | | | | |
| Input subsidies availability | √ | | | | | | |
| Guaranteed minimum price | √ | | | | | | |
| Increase in income generation | | | | √ | | | √ |
| Value addition | | | | | | | √ |
| Market | | | | | | | |
| Scale of market | √ | √ | √ | √ | √ | √ | √ |
| Market availability | √ | √ | √ | √ | √ | √ | √ |
| Labour availability | √ | √ | √ | √ | √ | √ | √ |
| Strength of supply chain | √ | √ | | √ | √ | √ | √ |
| State of road infrastructure | | | √ | √ | √ | | √ |
| Social | | | | | | | |
| Improving quality of life | √ | √ | √ | √ | √ | | √ |
| Local employment | √ | √ | √ | √ | | √ | √ |
| Improving food security | √ | √ | √ | √ | √ | √ | √ |
| Improving local livelihoods | | | √ | | | | √ |
| Reducing theft levels | | | | √ | | | |
| Capacity building | | | √ | √ | √ | √ | |
| Knowledge | | | | | | | |
| Extension/advisory support availability | √ | √ | √ | √ | √ | √ | √ |
| Level of confidence | √ | | √ | √ | √ | √ | √ |
| Understanding of farming tools/equipment | | √ | √ | √ | √ | √ | √ |
| State of farming knowledge | √ | | √ | √ | √ | √ | √ |
| Regulation | | | | | | | |
| Food safety | √ | √ | √ | √ | √ | √ | √ |
| Permissible crops | | | √ | | √ | | |
| Water abstraction | | | | √ | | √ | |
| Water rights | | | | √ | | | √ |
| Changing land use | | | | | | √ | √ |
| Health and safety | | | | | | | √ |
| Environment | | | | | | | |
| Crop rotation | √ | √ | √ | √ | √ | √ | √ |
| Tree planting | √ | √ | √ | √ | | | √ |
| Not burning crop residues in fields | √ | √ | √ | | | | |
| Hand weeding to reduce use of weed killers | | | | √ | | √ | √ |
| Soil fertility improvement using lime/manure/Sun hem/velvet beans | | √ | √ | √ | | √ | |
| Fallowing | | | | √ | | | |
| Use of biological control for pest control | | | | √ | | | |
| Soil erosion | √ | √ | √ | | √ | | |
| Loss of carbon sinks | √ | | | | | | |
| Loss of wind breakers (trees) | √ | √ | √ | | | | |
| Poor rainfall patterns | | | | | √ | | |

3.2.1. Market and Financial Subdomains Considered in Land-Use Trade-off Decision Making

Within the market domain, respondents were more concerned with making expansion decisions based on secure market for the product and scale of the market. Availability of market was considered to be an important factor in sustaining the farming activities and largely influenced the decision to expand. This was highlighted during the discussions.

> Large-scale individually owned—"*Market availability for our produce is very important because before we think of expanding crop production we firstly think of where we will sell our produce... If we find market then we will be able to sell our produce and expand our production.*"

Respondents considered the strength of the supply chain, in particular, the ability of input suppliers to make timely deliveries—especially in light of the COVID-19 pandemic's effects—as being important in the decision to expand.

> Shareholder-owned—"*Supply chain issues are more important now than ever due to the effects of the COVID–19 pandemic . . . raw materials have to be ordered a year ahead now . . . the supply chain, particularly the ability of input suppliers to make timely supplies has a huge influence on our decision to expand [the cropped area]*".

The need to adhere to certification requirements (e.g., the Global GAP certification scheme) to enter certain markets was also an important consideration in the land-use trade-off decision-making process. The requirements were mostly anchored on greening the crop production system. This was prominent in the shareholder ownership model.

In terms of the financial domain, participants were more concerned with making expansion decisions based on profitability of the produce and quick payback period. During discussions, profitable crop production with quick payback period was considered to increase the likelihood to expand.

> Large-scale individually owned—"*Quick payback period and profitability [of the produce] does matter a lot to us in the decision to expand [the cropped area] and it is on this basis that we choose which crop should be given high priority... It helps us to assess whether we will get good returns in the shortest time possible should we decide to expand.*"

Discussions with respondents revealed that availability of capital and access to credit, as well as variability of profits were considered in the decision to expand across most ownership models. It was noted that while availability of capital and access to credit increased the likelihood to expand the cropped area, variability of profits reduced the likelihood to expand the area under cultivation.

For the small-scale ownership model, participants were concerned with making expansion decisions based on availability of input subsidies through the country's Farmer Input Support Programme (FISP). During discussions, access to the input subsidies was considered to increase the likelihood to expand.

> Small-scale individually owned—"*The cost of inputs is too high, especially for maize . . . Availability of FISP [Farmer Input Support Programme] helps to reduce the cost of inputs which enables us to expand [the cropped area].*"

Other considerations in the decision to expand, in particular for the small-scale individual ownership model, were guaranteed minimum prices and income diversification arising from crop diversification. These factors were considered to increase the likelihood of expanding the area under cultivation.

3.2.2. Social Well-Being and Regulation Subdomains Considered in Land-Use Trade-off Decision Making

Within the social well-being domain, three factors were frequently mentioned as being important in the decision to expand across the ownership models. These were improving quality of life, increasing local employment and enhancing local food security. During

discussions, strong views emerged that improving quality of life, local employment and food security increased the likelihood of expansion. Looking across the ownership models, the collective and government ownership models tended to place a strong emphasis on improving quality of life and fostering food security. This was brought up during the discussions.

> Collectively owned—"*As a cooperative, our aim when deciding to expand [the cropped area] is to improve the quality of life of our members . . . this is very key for us.*"

> Government-owned—"*We think of improving food security when making expansion [of cropped area] decisions...For us producing more crops to foster food security influences the decision to expand [the area under cultivation].*"

For the collective ownership model, fostering capacity building in good farming practices through the exchange of ideas among members of the ownership model was an important factor that influenced the decision to expand. Discussions revealed that capacity building in good farming practices improved productivity which reduced the likelihood of expansion.

In terms of the regulation domain, an important factor that consistently emerged in the discussions across the ownership models was adherence to regulations on food safety. However, discussions with respondents suggested that awareness of regulations that influence the decision to expand was limited across the ownership models.

### 3.2.3. Knowledge Base and Environment Subdomains Considered in Land-Use Trade-off Decision Making

Within the knowledge base domain, three factors that were consistently highlighted as being important in the decision-making process across all ownership models included farming knowledge, understanding farming tools/equipment and availability of extension/advisory support services. Farming knowledge and availability of extension services in good farming practices were considered to reduce the likelihood of expansion.

> Medium-scale individually owned—"*Having extension services around that promote good farming practices helps to increase crop productivity . . . this helps to express the pressure to expand into forest area looking for fertile soils*".

> Large-scale individually owned—"*Farming knowledge is key . . . especially knowledge in good farming practices . . . we get more yield and this reduces chances of expanding [the cropped area].*"

In terms of the environment domain, respondents were concerned with various factors anchored on environmental stewardship and the effects of expanding agricultural areas into natural habitats on the environment. Discussions revealed that consideration of factors anchored on environmental stewardship reduced the likelihood of expansion, while those taking into account environmental impacts associated with expansion reduced the likelihood of expansion. With regards to environmental stewardship, factors that were highlighted included crop rotation, use of manure and planting of velvet beans and Sun hem to improve soil fertility. Consideration of these factors was prominent among government, shareholder, medium-scale and large-scale individual ownership models. During discussions, it was noted that improving soil fertility helped to enhance productivity which in turn reduced the likelihood to expand the cropped areas into natural habitats in search of fertile soil.

> Large-scale individually owned—"*We practice crop rotation in order to improve soil fertility of our farming fields . . . improved soil fertility improves our productivity which reduces chances of expanding into forest areas to look for fertile soils.*"

Other factors anchored on environmental stewardship included avoiding burning of crop residues in farming fields and undertaking hand weeding rather than using agrochemical weed-killers. During discussions, it was noted that these environmentally friendly practices helped to protect soil life which, in turn, reduced the likelihood of expansion into

forest areas for fertile soils. Consideration of hand weeding was prominent among the shareholder and government ownership models, while avoiding burning crop residues in agricultural fields was common among small, medium and large-scale ownership models. In terms of environmental impacts of expansion, factors mostly considered in the decision to expand included soil erosion and loss of wind breaks due to cutting down of trees during expansion. Consideration of these factors reduced the likelihood of expansion and was prominent among the collective, small, medium and large-scale ownership models.

## 4. Discussion

The findings of this study have shown that land-use trade-off decisions are influenced by multiple factors. The influence of multiple factors in the decision to expand was evident across all the farming ownership models considered in this study. This finding lends credence to earlier studies that highlight the influence of multiple factors in land-use decision making [11,13,47].

Regarding insights into factors that influence the decision to expand, the market domain was the most influential. This was evident across all the ownership models. This is consistent with the observation that market demand for agricultural products influences decisions involving forest conversion to agricultural land [48]. Results showed that market availability, scale of the market and strength of the supply chain are important considerations in the decision to expand. Our findings are in line with Journeaux et al. [11] who highlight the importance of market availability in land-use change decisions. Overall, our findings suggest that consumer demands for more green products and the private sector's commitments to green their supply chains can help to influence land-use decisions that are less likely to cause environmental harm. Our results also suggest that government regulations that improve market access for greener products can also help to influence land-use decisions that are less likely to cause environmental harm.

The financial domain was also given high consideration in the decision to expand. In particular, profitability, variability in profits and payback period are underlined as important factors in the decision to expand. This finding is in line with earlier research that emphasizes the significance of economic returns and variations in profitability in land-use change decision making [15,49]. It also aligns with those of Lubowski et al. [50], who found a correlation between loss in crop productivity and a decrease in cropland. These results suggest that financial regulations and strategies that promote profitable environmentally friendly products with fairly stable profit margins and a quick payback period can to help influence land-use decisions that are less likely to cause environmental harm.

Our findings also showed that availability of input subsidies through the government-supported Farm Input Support Programme (FISP) influenced the decision to expand, particularly for the small-scale individual ownership model. Under FISP, small scale farmers receive input subsidies which are aimed at improving agricultural productivity and reducing the cost of production [51,52]. Some authors have noted that agricultural policies that increase the relative returns to agriculture precipitate forest clearing for crop production [53,54]. This is consistent with Bulte et al. [55] who found a positive correlation between input subsidies and country-level deforestation. Our findings suggest that increasing agricultural productivity through input subsidies encourages the decision to expand. This suggests that promoting agricultural productivity through intensification may require accompanying policies and strategies to safeguard forest ecosystems from agricultural expansion, as noted by Adolph et al. [56].

Our results also showed that availability of capital and access to credit influenced the decision to expand. Past research notes that farmers can buy improved seed, fertilizer and agrochemicals that increase productivity when they have access to finance and credit [57]. It is also reported that having access to finance and credit encourages farmers to make investment in farming inputs that improve productivity [58,59]. Dong et al. [60] also found that eliminating credit constraints increased agricultural productivity. This is consistent with Omonona et al. [61] who observed that agricultural productivity was higher among

farmers who did not have credit constraints than those who did. Since our results showed that increasing agricultural productivity through access to credit and capital availability encourages agricultural expansion into the natural habitat, this finding suggests that those providing finance have the power to influence expansion decision-making processes and make them more environmentally friendly. For instance, they may require applications for credit and loans to be accompanied by an action plan for adopting environmentally friendly agricultural practices.

Past studies have shown that regulations are important in influencing land-use change decision-making processes [62,63]. However, our findings showed that the level of awareness of regulations that influences land-use trade-off decision making was low across the ownership models. This suggests the need to develop strategies that raise awareness on laws that encourage land-use trade-off options that are less likely to harm the environment. In terms of the influence of the social domain on the land-use trade-off decision making, our findings showed that improving quality of life, local employment and food security increased the likelihood to expand. This provides support for past research that found that social outcomes influence farmers' land-use decision making [64].

Looking across the ownership models, availability of extension/advisory services was an important consideration in the decision to expand. Discussions with participants revealed that availability of extension services in good farming practices helped to improve productivity while assisting farmers to decrease the detrimental effects on the environment. This is in line with Samaniego et al. [65] who found that offering extension/advisory services facilitated access to knowledge that supported agricultural productivity by giving farmers a wide range of essential technical information. Our results suggest that agricultural policies that foster farming knowledge and provision of extension services can help to influence land-use decisions that are less likely to cause environmental harm.

Our research found that while the environmental domain did not appear to be given high consideration in the decision to expand given its low weight (0.08), land managers across all the ownership models highlighted various environmental stewardship activities that reduced the likelihood to expand. Key activities included those tailored to improving soil fertility and maintaining soil life such as crop rotation and planting velvet beans and Sun hem, inter alia. This finding provides support for earlier work by Journeaux et al. [11] that emphasized the importance of soil fertility in influencing land-use change decisions.

Discussions with participants revealed that the detrimental effects of agricultural expansion on the environment reduced the likelihood to expand. Key impacts taken into account across the ownership models included soil erosion and loss of wind breakers due to cutting down of trees during expansion. This finding concurs with the observation by Malek et al. [48] that environmental implications are taken into account by decision-makers when making land-use change decisions. In spite of this, our findings have shown that land managers are more likely to trade-off environmental concerns when they are compared to other factors such as market and finances. This suggests the need to advance strategies that foster agricultural systems that adopt environmentally friendly practices.

It is evident from our study that while there are some commonalities in the factors considered by the agricultural land managers with distinct ownership models, some key differences are also evident. This suggests that ownership models do influence the factors that are taken into account. For instance, relative to the other ownership models, the government and collective ownership types had high considerations for social factors, whereas the NGO ownership type prioritized environmental aspects. Therefore, in addition to devising policies and actions to influence common decision-making factors, there is need to advance tailored strategies that address differences in priorities and decision making emanating from variations in farming business ownership models.

## 5. Conclusions

To better manage competing land-use objectives concerning agricultural production and forest conservation, understanding factors that influence land-use trade-offs is funda-

mental. This is critical given that managing these trade-offs remains poorly understood in African settings, notably Zambia. This study applied elements of the AHP methodology, an MCDM approach, to test its applicability using distinct farming business ownership models in Zambia. The study aimed to gain insights into factors that agricultural land managers with distinct ownership models consider in land-use trade-off decision making involving agricultural expansion into natural habitats.

Our study has shown that AHP is an applicable participatory approach that can provide insights into the relative importance of factors taken into account when agricultural land managers in Zambia make decisions on trade-offs in land usage. Our study has also shown the value of AHP as a tool for future research on land-use trade-off decision making, particularly where there are multiple and competing objectives of increasing agriculture production and forest conservation.

Important criteria in land-use trade-off decision making were identified in our study and classified into six higher-level domains, namely social, market, financial, environmental, knowledge and regulatory. These were included in the MCDM process. Our findings showed that land-use trade-off decision making was not completely dominated by one domain, although on average, the market domain was weighted more highly than others, followed by the financial domain. Results also showed that while the environmental domain appeared to have less influence on the decision-making process given its low weight, instances were highlighted where agricultural land managers prioritized environmentally friendly land-use decisions. Nevertheless, land managers were more ready to compromise on environmental concerns when these were weighted against other criteria such as market and finances. This underlines the requirement to support policies that encourage the adoption of environmentally friendly practices.

The findings of our study have shown that especially for the small-scale individual ownership model, the availability of input subsidies through FISP influences the decision to expand. This implies that attempts to increase agricultural productivity through intensification may require accompanying measures to safeguard forest ecosystems from agricultural expansion.

Our study revealed that some key differences were evident in the factors that influenced the decision to expand among the distinct farming business ownership models. This showed that ownership models do have an impact on factors that are taken into account. This suggests the need for tailored strategies that address differences in priorities and decision making associated with distinct farming business ownership models.

The AHP methodology was applied with participants at the main domain level and weights generated in terms of overall importance of each of the domains. Therefore, in order to enhance our understanding of the applicability of the AHP methodology with distinct farming business ownership models, future research can extend the application of the methodology to the subdomains (criteria). This will help to highlight finer details of the land-use trade-off decision-making process in terms of the importance distinct farming business ownership models place on particular criteria within the main domain.

Our study engaged a small number of agricultural land managers with distinct farming business ownership models due to time and resource constraints. Therefore, we proceed with caution in drawing any specific policy recommendations. Nevertheless, our study is useful in providing insights into land-use trade-off decision-making processes involving distinct farming business ownership models. Overall, the results of our study have shown that managing land-use trade-offs for competing land-use objectives involving agricultural production and forest conservation requires an understanding of how those who own and manage such land consider and manage those challenging land-use trade-offs. Such an understanding can help provide insights on how to negotiate the trade-offs and influence those decision-making processes.

**Author Contributions:** Conceptualization, J.M.K. and X.H.J.; Methodology, J.M.K., X.H.J. and A.R.; Data analysis, J.M.K.; Original manuscript, J.M.K.; Project administration, J.M.K. and X.H.J.; Writing–review and editing, J.M.K., X.H.J. and A.R.; Validation, J.M.K., X.H.J. and A.R. All authors have read and agreed to the published version of the manuscript.

**Funding:** Funding for this study was provided by UK Research and Innovation through the Global Challenges Research Fund programme, "Growing research capability to meet the challenges faced by developing countries". This is through the Sentinel Project—Social and Environmental Trade-offs in African Agriculture—with the grant number being ES/P011306/1. The AHP model development was funded by the New Zealand Ministry for Business, Innovation and Employment's Our Land and Water National Science Challenge (contract C10X1507) as part of the Next Generation Systems Programme.

**Data Availability Statement:** Data presented in this study can be obtained from the corresponding author on request. As a result of the continuity of the research study, data are not publicly available.

**Acknowledgments:** We would like to thank the technical staff from Ministry of Agriculture in Katete, Chipata, Chongwe and Mkushi districts for their support in linking us to the agricultural land managers (farmers). We are also grateful to the agricultural land managers in Katete, Chipata, Chongwe and Mkushi for providing the data used in this study. We also thank Darius Phiri for developing the study site map. We also wish to express our gratitude to Jo Davies, Nugun Patrick Jellason and Beth Downe for their useful comments that helped to improve the quality of the manuscript.

**Conflicts of Interest:** The authors have no conflict of interest to declare.

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
