# Peer review of "Understanding Land-Use Trade-off Decision Making Using the Analytical Hierarchy Process: Insights from Agricultural Land Managers in Zambia"

_land, doi:10.3390/land12030532_

Round 1

Reviewer 1 Report

i) After using the AHP, it is necessary to display the pairwise comparison matrix of weighted factors, perform a consistency check, and present a list of subcriteria weights. However, this step is a hindrance in the manuscript. The current presentation of the AHP is inadequate.

ii) In your literature review and discussions, focus on highlighting the strength of AHP in making land use decisions, particularly when various stakeholders have conflicting interests. Start with a compelling story, but also firmly support your methodology and explain how it adds to, confirms or contradicts previous research. There are numerous studies that utilized AHP to determine the best land use. Your literature review lacks a comparison style, merely presenting important factors and benefits of AHP.Recommended readings:

https://doi.org/10.3389/frwa.2020.579087

https://doi.org/10.1007/s41742-021-00326-0

https://doi.org/10.3390/su9122196

iii) Your discussions should consider incorporating land use inventories. Given your focus on trade-offs, countries, particularly in Africa, require official, registered records of land use and cover to effectively manage their land use. Your results can play a role in creating and updating these inventories. Recommended readings:

https://doi.org/10.1080/00396265.2017.1383711

https://doi.org/10.1065/lca2007.10.364

https://doi.org/10.1007/s10661-015-5056-7

Reviewer 2 Report

This paper innovatively implemented one of the established MCDA techniques - AHP to gain insights into factors that agricultural land managers with distinct ownership models consider in land-use trade-off decision-making involving agricultural expansion into natural habitats. The paper is well structured. The technical quality of the paper is good, as well as the results obtained. I appreciate the interviewing process while developing the AHP analytics. The publication is recommended, provided the following corrections are performed.

Currently, there are many researchers who successfully implemented machine and deep learning techniques for describing the non-linear interrelationships between the MCDA attributes/domains which AHP often overlooks. Describe the relevance of implementing AHP with reference to the non-linear nature of land-use trade-offs and contemporary data science research.

There are some errors in “Tense and Grammar” throughout the manuscript. Therefore, diligent editing is required to fix the ENGLISH language.

Provide a high-resolution study area map for better readability.

Figure2. –  follow standard flow diagram symbols and rules.

Improve the quality of all the graphs in the manuscript. Authors are encouraged to use the python graph development library of some professional graph development tools. Also, use a consistent font type and size in all the graphs presented in the manuscript.

Reviewer 3 Report

General comments

The study is impressive in the context of visual quality assessment of the specific area. Certainly, such analyses help to formulate policies for better land use trade off decisions of the adjacent regions as well. Analytical Hierarchy to distinct farming business ownership models in Zambia  makes it an attractive manuscript to be published. After addressing the minor comments the manuscript shall be published.

Specific comments

In the abstract, mention the most significant results instead of the General conclusion of the article.

The use of punctuation and lengthy sentences must be revised majorly in the article.

Please support your discussion section through citations of previous relevant and latest studies.

The conclusion portion should be elaborated as the the conclusion of a research paper has several key elements you should make sure to include:

Ø  A restatement of the research problem

Ø  A summary of your key arguments and/or findings

Ø  A short discussion of the implications of your research

Round 2

Reviewer 1 Report

I have no further comments. Thank you.